# Verbal Interactional Synchronization between Therapist and Children with Autism Spectrum Disorder during Dolphin Assisted Therapy: Five Case Studies

**DOI:** 10.3390/ani9100716

**Published:** 2019-09-24

**Authors:** Richard Griffioen, Steffie van der Steen, Ralf F. A. Cox, Theo Verheggen, Marie-Jose Enders-Slegers

**Affiliations:** 1Department of Psychology and Education, Open University of the Netherlands, 6419 AT Heerlen, The Netherlands; theoverheggen@gmail.com (T.V.); Marie-Jose.Enders@ou.nl (M.-J.E.-S.); 2Department of Special Needs Education and Youth Care, University of Groningen, 9712 TS Groningen, The Netherlands; s.van.der.steen@rug.nl; 3Department of Psychology, University of Groningen, 9712 TS Groningen, The Netherlands; r.f.a.cox@rug.nl

**Keywords:** animal assisted interventions, autism spectrum disorder, dolphin assisted therapy, verbal synchrony, turn-taking behavior

## Abstract

**Simple Summary:**

This study investigates the synchrony in conversations (i.e., turn-taking) between a therapist and five children with Autism Spectrum Disorder during dolphin-assisted therapy. Videos of the first and last dolphin-assisted therapy sessions were analyzed with regard to turn-taking between the therapist and child in the presence of a dolphin. The results show that adequate turn-taking seemed to increase over time, but mainly for children who had reasonable verbal communication skills at the start of the therapy sessions.

**Abstract:**

Synchronizing behaviors in interactions, such as during turn-taking, are often impaired in children with Autism Spectrum Disorder. Therapies that focus on turn-taking generally lead to increased social skills, less interruptions, and silent pauses, however a positive non-demanding environment is therefore thought to be beneficial. Such an environment can be achieved by incorporating animals into therapy. Our study was guided by the following research questions: (1) How can we characterize the interaction between child and therapist during dolphin-assisted therapy, with regard to synchrony in verbalizations (turn-taking) and (2) does synchrony change over the course of six sessions of therapy? To answer these questions, we performed a cross-recurrence quantification analysis on behavioral data of five children, to give a detailed view of the interaction between therapist and child in the context of dolphin-assisted therapy. We were able to detect synchrony (i.e., adequate turn-taking) in all dyads, although not all children improved equally. The differences might be explained by a delayed reaction time of some children, and their level of language development.

## 1. Introduction

Children that are diagnosed with Autism Spectrum Disorder (ASD) have limitations with regard to their functional and effective communication and are impaired in initiating and sustaining reciprocal social interactions [1]. While there is large variation in the language and communication skills of children with ASD, there is often a limitation in the ability to adapt their language to the interaction partner, the social context, and the situation [2,3,4]. Deficits in social skills, such as play and imitation, are also highly prevalent in children with ASD [5,6]. Furthermore, the social and communication problems that these children experience are often not compensated for by the use of gestures or eye contact [5,7].

An important part of social interactions is synchrony, a rhythmic pattern of behavior that is mutually regulated, reciprocal, and harmonious [8,9,10,11]. Synchrony emerges between two interaction partners [12,13,14], and it is expressed as a temporal match between their behaviors [8,15]. Synchrony, in terms of speed, rhythm, and amplitude of verbalizations and movements, can increase the social attunement between interaction partners, and can lead to a higher quality of social interactions [14,16]. In this way, synchrony in interactions facilitates the development of social emotional skills [8,9,17,18,19]. Recent studies suggest that there is less synchronization in the interactions of children with ASD [15,20,21,22,23]. Children with ASD have trouble in achieving mutual regulation and temporal coordination in their interactions with others and, as a consequence, it is hard for them to develop their social emotional skills [8,9]. Therefore, several experts point out the importance of specific studies on synchrony-related interventions for children with ASD [15,24].

Synchrony in verbal interactions is a complex behavioral pattern, in which one speaker anticipates what the other speaker does, and vice versa. It is characterized by adequate turn-taking behavior, that is, alternating verbalizations of varying size between two or more persons [25,26,27,28]. Similar to other synchronizing behaviors, turn-taking behavior is often impaired in children with ASD [15]. Therefore, some researchers suggest that improving the turn-taking behavior leads to fewer interruptions and silent pauses and increased social skills [16,29,30,31]. It is important to be aware of the role of the parents (or therapist) when it comes to turn-taking, and especially their sensitivity to the child. For example, the study of Siller and Sigman [29] shows that parents of children with ASD who show higher levels of synchronization stimulate their children’s joint attention and language skills to a greater extent than parents who show less synchronization. However, at the same time, a recent study shows that a more directive style of parents to achieve synchrony, such as placing more demands on their child’s attentional focus, is associated with a negative affect of young children who are at risk of an ASD diagnosis [30]. In other words, trying to impose synchronous behavior in children with ASD might sometimes be counterproductive. Therapies that increase synchrony, but at the same time create a positive non-demanding environment, are therefore thought to be beneficial.

Some authors argue that such a positive and non-demanding environment can be achieved by incorporating animals into therapy and that animals would contribute to children’s social attunement [30]. Animals can have a positive effect on our psychological and physical well-being, and this positively influences our social interactions and helps us to regulate our emotions [32,33]. Various studies have shown that Animal Assisted Therapy (AAT) can be promising in itself, as well as an addition to existing interventions [34,35,36,37,38,39]. Researchers have indicated that the presence of animals, such as dolphins, might make children with ASD feel safe in a therapy environment [40]. Research specifically focusing on dolphin-assisted therapy shows that this intervention positively contributes to the language development of children with ASD [34,41,42], and that dolphin-assisted therapy has a positive effect on social interaction and greater self-esteem [34,41,43,44]. Studies on interventions with other animals report decreased behavioral problems [38], increased social interaction and communication [45], a more playful mood, more focus, and more awareness of the social environment [46].

In summary, synchronizing behaviors, such as turn-taking, are often impaired in children with ASD [15]. Improving turn-taking behavior would lead to improved social interaction, less interruptions, and silent pauses [16,29,30,31]. Therapies that aim to increase synchrony are therefore valuable, but a positive non-demanding environment seems to be thought as beneficial [30]. Such an environment can be achieved by incorporating animals into therapy [43]. While most of the studies on AAT report increased social and communication skills of children with ASD, the effect of this therapy on synchronizing behaviors, and especially turn-taking has not been investigated.

Our study was guided by the following research question: How can we characterize the interaction between child and therapist during dolphin-assisted therapy, with regard to synchrony in verbalizations (turn-taking), and does synchrony change over the course of six sessions of therapy? To answer these questions, we performed a cross-recurrence quantification analysis (see methods section), to give a detailed view of the shared dynamics between the therapist and child in the context of dolphin-assisted therapy.

## 2. Materials and Methods

### 2.1. Ethical Considerations

In this study, five children with ASD were followed for six weeks while they received dolphin-assisted therapy. At the start of the study, written consent was obtained from the parents of the children. The parents were also informed about the aim of the study, as well as about the protocol of the therapy. It was made clear to them that they were free to stop their participation at any moment. During the research, the dolphins’ welfare was continually monitored by a veterinarian. All of the necessary precautions were taken to prevent possible harm to the dolphins, such as limiting the time of interacting with people to a minimum and not allowing people to be in the water (swim) with the dolphins. The study was evaluated by the Health Care Inspectorate (IGZ) of the Dutch Ministry of Health, Welfare, and Sport, and assessed as a low-risk study.

### 2.2. Participants

Five children (four males, one female) with ASD participated in this study. The children were recruited through the website of the SAM Foundation, (www.stichtingsam.nl), which facilitated dolphin-assisted therapy (DAT) in the Netherlands. Parents voluntarily registrated their children for therapy after reading newspaper articles about this program. Upon registration, the parents were asked for permission to participate in the study. After registration, the children were placed on a waitlist and were called in for an extensive intake as soon as there was room to start therapy. The exclusion criteria for participation included fear of water, epilepsy, poor vision, and a history of aggressive behavior toward animals. The therapy program was partly sponsored, and therefore parents paid a reduced price.

We obtained information regarding the speech language development of the children from intake forms that were completed by the parents at the start of the therapy program, such as the number of words that they used and whether they were able to express their feelings. The parents of three children indicated that they only used single words to communicate, and the parents of the remaining two children indicated they could produce short sentences (3–4 words). Table 1 provides a detailed description of the participants.

### 2.3. Procedure

The therapy took place in the Netherlands at the Dolphinarium Harderwijk. The dolphins live in an artificial lagune of 7000 square meters and are not used for shows. They live together as a small pod (17 dolphins, including males, females, juveniles, and babies). The SAM foundation was responsible for the organization of the program. They provided the trained speech and language therapists. All therapists were trained at universities of applied science in the Netherlands and had followed an intensive training on how to work with the dolphins, organized by the foundation.

This professional team, consisting of three female therapists and three female dolphin trainers, guided the children through this specific DAT program. Each child worked with one therapist and one dolphin trainer during six sessions. All of the therapists had at least ten years of experience in working with children with disabilities. The therapy setting with the child, therapist, dolphin trainer, and dolphin was set up to create an atmosphere in which the child felt safe [40,47,48]. According to the literature, children feel attracted by the dolphin because of its anatomically shaped smile and its gentle movements in the water [42]. The interaction between child, dolphin, and therapist enables triangular communication (transmission triangle or carry-over) to provoke children’s verbal or non-verbal responses [34]. This means that the interaction between dolphin, therapist, and child creates a triangle in which a redirection of communication via the animal is established. While exercising turn-taking behavior, the therapist teaches the child how to act toward the dolphin and in return the child will respond with attention to the therapist question.

That is, observing the animal’s behavior, interacting with the animal, and watching other people interact with the animal can stimulate children’s social and communication skills.

We worked with three dolphins for five children, and each child worked with the same dolphin during the sessions. At the start of a session, the dolphin trainer would call the dolphins to the raft, and one dolphin was chosen to work with during the session. After the first session, the preferred choice of dolphin was always the dolphin the child had worked with before. Due to pregnancy of one dolphin, we had to change dolphins for two children in the last session. In addition, because the dolphins were not forced to interact with the children, we only once switched dolphins when the regular dolphin did not come to the raft when the dolphin trainer called. We did not have to cancel a session due to stress signals of the dolphins. In an earlier study, the stress signals of the dolphins were observed before, during, and after therapy sessions [34]. In this study, the respiratory frequency of the dolphins was measured as a parameter of arousal. Observations were made by means of a digital camera and by an observer, who noted the breathing frequency and sounds of the dolphins. The videotape was watched afterward and all of the behaviors were recorded in an ethogram. No significant differences between therapy sessions and control sessions were found.

The dolphin trainer selected the dolphin at the start of the therapy sessions (depending on the dolphin’s condition and behavior at that moment), and supervised the behavior and possible stress signals of the dolphin. Before the therapist and child wanted to engage with the dolphin, they always had to ask for permission from the dolphin trainer and no confusing situations for the dolphin would occur. Apart from that, the dolphin trainers were instructed not to interfere in the interaction between child and dolphin or therapist, unless the situation called for immediate action (e.g., when the dolphin showed stress signals or inappropriate behavior).

During the six sessions, the child, therapist, and dolphin trainer were standing or sitting on a raft at the waterfront. At the start of the session, the therapist introduced the child to the dolphin and the dolphin-trainer. Subsequently, the therapist started a conversation about the dolphin and its behavior, inviting the child to verbally express his/her observations. The children were then taught to use gestures to make the dolphin perform certain actions. For example, children learned to signal, so that the dolphin would jump or clap with the fins. The child was only allowed for making the gestures if he/she first made eye contact and verbally asked permission. In this way, the children’s interactions with the therapist were reinforced by their wish to play with the dolphin. Furthermore, to help the children express their emotions, 40 cm square boards with emotion pictures were shown, and the therapist started a conversation about these emotions. As recognizing, naming, and describing emotions is something that children with ASD often have trouble with, the therapy program incorporated a task in which these skills are practiced. To include the dolphin in this task, the boards were thrown into the water for the dolphin to return.

### 2.4. Measurements

#### Coding of Verbal Behavior

Videos of the first and last session of the therapy were systematically observed by means of timeserial event coding, while using the program MediaCoder, University of Groningen, Groningen, The Netherlands (a proprietary transcoding program for Microsoft Windows) [49]. This means that both the category of the behavior as well as the timing of that behavior were registered. We coded the following four behavioral categories: (1) Therapist speaks, (2) Therapist is silent, (3) Child speaks, and (4) Child is silent. Three student-raters first completed a training in which they coded one therapy session, and compared their codes with those of an expert-rater, who constructed the codebook with coding rules. Inter-rater reliability was considered to be sufficient when at least 80% of the codes of the rater and expert-rater were similar with regard to both the timing and the chosen category. If this percentage was not reached, the raters received an additional explanation of the coding rules and coded a second therapy session, after which the percentage of agreement was determined again. All of the raters reached sufficient inter-rater reliability (>80%) after coding two sessions and proceeded with the coding of the videos.

### 2.5. Data Analysis

We separately present the data for each participant. We first transformed the codes and accompanying times to a time series with a sampling rate of 2 Hz and then applied Cross Recurrence Quantification Analysis (CRQA; see [50,51,52]. CRQA originates from the natural sciences, and it has recently been introduced to the study of human interactions (see e.g., [53,54,55,56,57], and recently also human-animal interactions [58]. This study [58] investigated the course of behavioral synchrony on movement patterns (and the outcome on problem behaviors) of children with Down syndrome (DS) and children with ASD while interacting with a dog. We used CRQA to operationalize the movement synchrony between a dog and child and found an increase in the coupling between child and dog during the final session, which means that the child and dog became mutually attuned in their movements.

CRQA focuses on the shared dynamics of two coupled systems by detecting repeatedly occurring matches between the behavior of two interaction partners [59]. In this case, matching behavior was defined as correct turn-taking behavior in the conversation between child and therapist, that is, speaking when the other person is silent, and vice versa [16,60]. CRQA detects these matches across all possible timescales, which range from seconds to the duration of the entire interaction [59]. This means that, even when the appropriate turn is (slightly) delayed, it will still be detected.

It is possible to calculate several measures from CRQA. We looked at the Diagonal Cross-Recurrence Profile (DCRP; see e.g., [59,61,62,63]), allowing for leader-follower analysis of the interaction. Figure 1 shows the average DCRP of the first session for all five children, as an example and to highlight the main measures. Concretely, we focused on matching behaviors (adequate turn-taking) within an interval of 30 s around the Line of Synchrony (LOS). The LOS represents instances of matching behavior of child and therapist at the exact same time. This either means that the child speaks while the therapist remains silent, or vice versa. The percentage of matching behavior at the exact same moment (i.e., on the LOS) is called the percentage of synchrony (% Sync). Yet, the patterns of matching behavior do not always occur at the exactly same moment, there is often some delay [28]. Therefore the recurrence rate (RR) represents the proportion of matching behavior within an interval of 30 s around the LOS (15 s before/after the LOS). The highest proportion of matching behaviors that can be measured within that interval is called RR_peak_. A final measure that can be derived from the DCRP is Q_los_, which is the proportion of matching behavior on the left side of the LOS divided by the proportion of matching behavior on the right side. If Q_los_ is higher than 1, the therapist temporally leads the interaction. This means that the behavior of the therapist (speaking or listening) influences the child’s behavior to a greater extent than vice versa. If Q_los_ is smaller than 1, then the child temporally leads the interaction.

The Q_los_ measure can be calculated by dividing the proportion of recurrence on the left side of the LOS by the proportion of recurrence on the right side.

## 3. Results

Table 2 shows descriptive statistics rlatad to the CRQA analysis. Below, the results will be separately discussed for each participant.

### 3.1. Participant 1

According to her parents, participant 1 has good verbal skills. She speaks a lot, but does make grammatical errors. Formulating a coherent sentence takes more time, and she has difficulty expressing and interpreting emotions and feelings. She feels comfortable around animals.

#### Analysis of Turn-Taking

During the first session, we recorded 232 s total time of spoken language, of the participant and 542 s of spoken language of the therapist. During the last session, the seconds of spoken language of the participant increased to 278 (19.8%) and decreased for the therapist to 490 (9.8%). Figure 2 shows the Diagonal Cross-Recurrence Profile (DCRP) of participant 1 during the first and final session. The recurrence rate increased from 0.18 to 0.26. This means that the proportion of behavioral matches (i.e., adequate turn-taking) increased with 44.4%. The RR_peak_ measure, which is the highest proportion of recurrence observed, also showed an increase, from 0.24 to 0.29. The Q_los_ measures show that, during both sessions, the child more often temporally led the conversation than the therapist (both values < 1). When compared to the first session (Q_los_ = −0.03), the dominance of the child’s behavior in the interaction appeared to be slightly less (Q_los_ = −0.02).

### 3.2. Participant 2

As reported by his parents, participant 2 started speaking when he was three years old. His language development is delayed and he has difficulty producing meaningful sentences and expressing his emotions.

#### Analysis of Turn-Taking

During the first session, we recorded 199 s of spoken language of the participant and 599 s of the therapist. During the last session, the number of participants seconds of spoken language decreased to 138.5 (30.4%) and decreased for the therapist to 450 s (24.8%). The Diagonal Cross-Recurrence Profile (Figure 3) shows the recurrence rate, the proportion of behavioral matches (i.e., adequate turn-taking), slightly increased from 0.17 to 0.19. The highest proportion of recurrence (RR_peak_) increased from 0.23 to 0.24. The Q_los_ changed from −0.066 to −0.032. This means that the child temporally led the interaction, but that his dominance became less apparent over time. Overall, one could conclude that the conversational dynamics in terms of these measures did not significantly change between the two sessions.

### 3.3. Participant 3

According to his parents, participant 3 is able to speak, but it is hard for him to communicate with the right words. He has weak mouth motor skills, which results in difficulty to formulate sounds with his mouth and slow speech. He uses single words and sometimes 3–4-word sentences, with regular echolalia. He is quickly over-stimulated and is only able to communicate with supportive gestures. His parents often let him finish sentences, such as: “We go to” “the dolphins”.

#### Analysis of Turn-Taking

This participant’s total time of spoken language increased from 113.5 to 126 s (11.0%). Additionally, for the therapist, this increased from 562.5 to 605.5 s (7.6%). As Figure 4 shows, the recurrence rate remained relatively stable (a slight increase from 0.08 to 0.09). RR_peak_ also remained stable (a slight increase from 0.11 to 0.12). Q_los_ increased from 0.005 to 0.02, which means that the therapist temporally led the interaction, and even more so during the final session. As for participant 3, also for this child, the conversational dynamics in terms of these measures remained relatively stationary.

### 3.4. Participant 4

Participant 4 speaks unclearly according to his parents and rather points to what he wants. He speaks with single and simple words, or very short sentences in combination with gestures.

#### Analysis of Turn-Taking

There is a minor decrease in the participant’s total time of spoken language from 176 to 174.5 s (0.8%). The therapist’s total time of spoken language increased from 854 to 943 s (10.4%). The DCRP of participant 4 (Figure 5) showed lower RR_peak_ and RR_los_ during the final session as compared to the first session, with RR_peak_ decreasing from 0.19 to 0.07 and RR_los_ decreasing from 0.06 to 0.04. Q_los_ changed from −0.02 during the first session to −0.01 during the last session. This means that the child temporally led the interaction, but that his dominance became less apparent over time.

### 3.5. Participant 5

The fifth participant has trouble making certain sounds and is therefore not always well understood, but has a good vocabulary according to his parents. He uses short sentences of three to four words and supports this with gestures.

#### Analysis of Turn-Taking

We recorded 153 s of spoken language of the participant and 509 s of the therapist during the first session. During the last session, the number of participant’s seconds decreased to 34.5 (77.5%) and also for the therapist to 276.5 (45.6%). Figure 6 shows the DCRP of this participant during the first and final session. The RR_peak_ and RR_los_ decrease during the final session, respectively, from 0.2 to 0.08, and from 0.16 to 0.04, respectively. The Q_los_ is higher during the final session and it changes from negative to positive (from −0.04 to 0.04), which means that, during the first session, the child temporally led the interaction, while the therapist followed, whereas the therapist temporally led during the final session.

## 4. Discussion

Therefore, a positive non-demanding environment is thought to be beneficial to increase synchronizing verbal behavior, such as turn-taking, for children with ASD [29,30,31]. Such an environment can be created by incorporating animals into therapy situations [34,43]. However, until now, studies have not investigated whether synchronizing behaviors, and especially turn-taking, increase in the context of animal-assisted therapy (AAT). Therefore, this study examined videos of the interaction between child and therapist during dolphin-assisted therapy (DAT) in terms of turn-taking, while using data of five child-therapist interactions during both the first and final (sixth) session of DAT.

Because this exploratory study intended to investigate whether DAT influences changes in turn-taking behavior, it was beyond our scope to compare the effect of the therapy to a control group. However, in an earlier study [34], we reported the positive effect of this program on verbalization and social interaction measures, by comparing DAT to two control groups (using a radio-controlled boat in a swimming pool, and a waitlist). As advised by Marino [64] we minimized construct confounding, as we exposed the control group and the DAT intervention group to highly similar procedures.

However, the results of this study seemed only partially in line with our expectations that synchronization of verbalizations (i.e., turn-taking behavior) would improve during DAT. The data revealed increases in turn-taking for participants 1 and 2. They showed an increase in RR_peak_ and RR_los_, which implies that the proportion of “matching behaviors” (i.e., speaking while the other person remains silent, and vice versa) between the participant and the therapist increased from the first to the last session. The Q_los_ measures for participants 1 and 2 indicate that these children temporally led the interaction during both the first and last session, but that they became less dominant leaders during the final session. However, participant 3 showed no changes in RR_peak_ between the first and the last DAT session. The Q_los_ measures indicate that the therapist temporally led the interaction, and even more so during the final session. The final two participants (4 and 5) showed a decrease in synchronization of verbalization (verbal turn-taking behavior). Their RR_peak_ and RR_los_ measures decreased between the first and the sixth DAT session. The Q_los_ measures revealed that participant 4 temporally led the interaction during the first session, but this became less apparent during the final session. Participant 5 also led the interaction during the first session, but this was reversed in the final session, when the therapist mostly led the interaction.

For three out of five children, the total time of spoken language increased during the final session of DAT (see Table 2), while the total time of spoken language of the therapist measured in seconds decreased.

Some authors have suggested that turn-taking behavior requires a reasonable good vocabulary [31]. Taking an active role both as speaker-initiator or listener requires sufficient communication skills while children with disabilities mostly lack such skills [65]. Therefore, some differences in our results can be explained by initial difference in communication skills between the children. While participants 1 and 2 had less trouble speaking, the parents of participants 3, 4, and 5 reported that their children were only able to speak while using supportive gestures (i.e., substituting words for gestures, frequent use of pointing instead of verbal indications). Participants 1 and 2, with better communication skills, indeed seemed to improve with regard to their turn-taking skills, which confirms the suggestion of Yoder and Stone [31] that a reasonable vocabulary is a prerequisite for adequate turn-taking. Future studies on animal-assisted interventions and turn-taking are therefore advised to take a baseline level of verbal skills into account, e.g., using videotapes of the participants prior to the study.

To our knowledge, this study is the first to investigate the effect of DAT on verbal synchronizing behaviors (turn-taking). While a bigger sample would aid in the generalization of our findings, our small sample size enabled us to perform an in-depth analysis. The most important result revealed in this study, that is, turn-taking during DAT seems to increase for children with reasonable verbal communication skills, could be further examined in larger studies, which could also include other variables, such as the age of the children, the severity of the disorder, or gender.

Note that the five children in this study worked with three different therapists, and that the dolphin changed depending on its condition and availability. Although the same therapist always guided each child during the sessions, differences in the therapists’ behaviors might affect children’s synchronizing behaviors and turn-taking skills. Given that this was an exploratory study (do we see any changes in synchrony?), and because of the labour-intensive nature of the coding, this study only compares the child-therapist synchrony in two sessions. Future research could investigate more sessions to track the development of therapist-child synchrony over time. Moreover, differences in the dolphin’s behavior could have influenced the communication between the therapist and child, and the child’s experience of the therapy setting. Although this might be hard to incorporate in practice, future studies are advised to let the same therapist guide all children, and to use the same dolphin throughout the sessions. Alternatively, therapists could receive more extensive instructions with regard to what they say to the participants and instructions with regard to taking delayed reaction times into account.

There is an indication that children with ASD show additional attention deficits and a relatively delayed reaction time when compared to typically developing children. This may result in the need for more time to verbally respond to others [66,67]. These attentional deficits and delayed reaction time were not taken into account in this study, but may have influenced the results on turn-taking behavior. For example, when a question takes more time to answer for the child, and the therapist is already moving on, a turn is “missed”. According to Levinson [26], “turn-taking precedes language in ontogeny, but when language is acquired children struggle for years to squeeze complex language into short turn sizes with adult response time.” This skill might require even more time to master for children with ASD.

In future studies, the effect of Animal-Assisted interventions on synchrony could be further investigated, while using conditions of therapy with and without animals, or by comparing synchrony between child and therapist in the presence of different animals, such as dogs or horses. This will allow us to check the synchrony of turn-taking behavior with and without an animal and compare the different species. The results of this research can be useful in future research, because it makes clear that there is a great variation in language development in children with ASD. Finally, future studies could also investigate the optimal duration of DAT programs. It could be that a longer DAT program yields better results for children with weaker verbalization skills.

## Figures and Tables

**Figure 1 animals-09-00716-f001:**
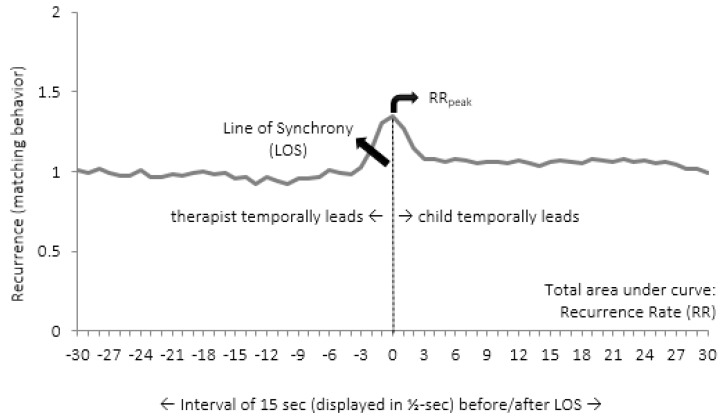
Average Diagonal Cross-Recurrence Profile (DCRP) plot of the first session, with an indication of the Line of Synchrony (LOS), Recurrence Rate (RR), and RR_peak_.

**Figure 2 animals-09-00716-f002:**
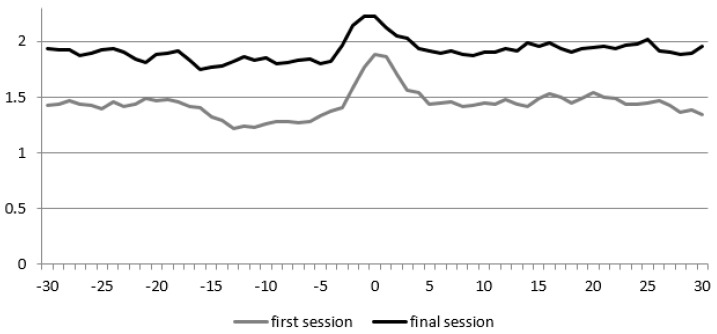
Diagonal Cross-Recurrence Profile (DCRP) of participant 1. The X-axis displays the delay in steps of 0.5 s, and the Y-axis the recurrence rate (i.e., the proportion of accurate turn-taking).

**Figure 3 animals-09-00716-f003:**
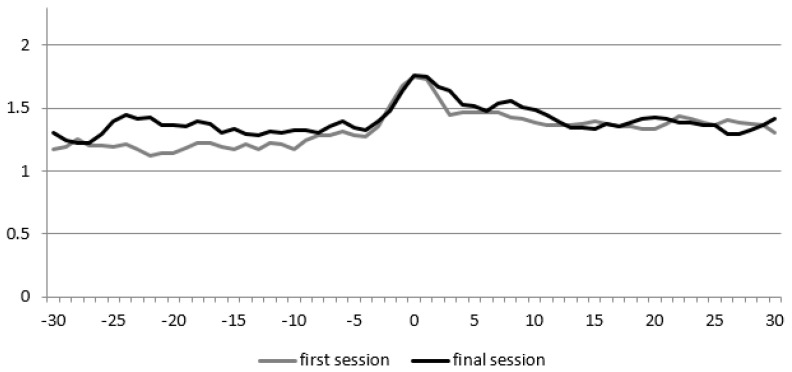
Diagonal Cross-Recurrence Profile (DCRP) of participant 2. The X-axis displays the delay in steps of 0.5 s, and the Y-axis the recurrence rate (the proportion of accurate turn-taking).

**Figure 4 animals-09-00716-f004:**
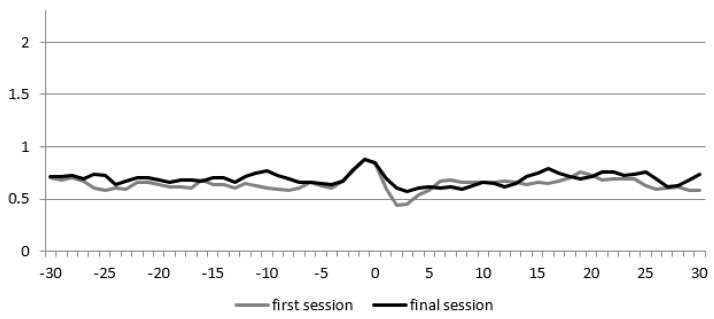
Diagonal Cross-Recurrence Profile (DCRP) of participant 3. The X-axis displays the delay in steps of 0.5 s, and the Y-axis the recurrence rate (the proportion of accurate turn-taking).

**Figure 5 animals-09-00716-f005:**
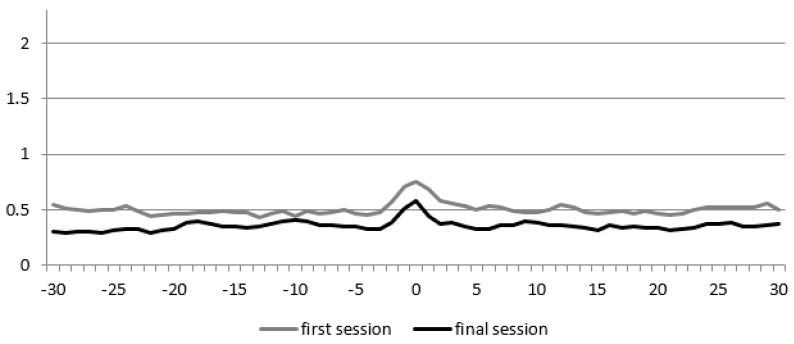
Diagonal Cross-Recurrence Profile (DCRP) of participant 4. The X-axis displays the delay in 0.5 s, and the Y-axis the recurrence rate (the proportion of accurate turn-taking).

**Figure 6 animals-09-00716-f006:**
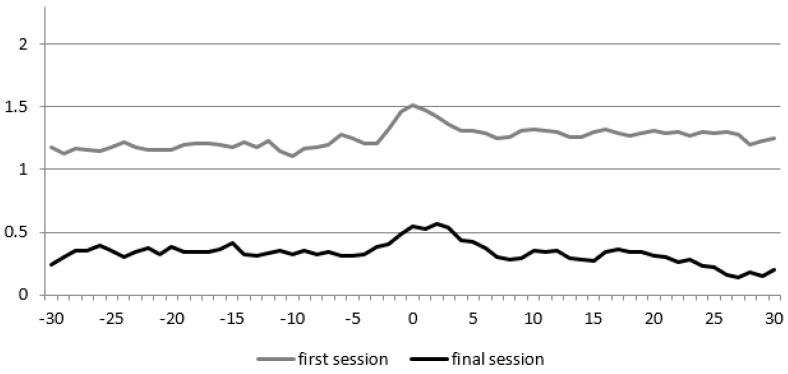
Diagonal Cross-Recurrence Profile (DCRP) of participant 5. The X-axis displays the delay in steps of 0.5 s, and the Y-axis the recurrence rate (the proportion of accurate turn-taking).

**Table 1 animals-09-00716-t001:** Participant characteristics.

Participant Characteristic	Pp 1	Pp 2	Pp 3	Pp 4	Pp 5
Age	7.5	8.5	6	7.5	8
Gender	Female	Male	Male	Male	Male
Diagnosis	ASD	ASD/ADHD	ASD	ASD	ASD
Education	Special	Special	Special	Special	Special
Medical treatment	-	Salbutamol (Asthma) and Methylphenidate (ADHD)	-	Levetiracetam (Epilepsy)	-
Sensory problems	-	-	-	Glasses	Glasses
Motor problems	Physiotherapy	-	-	Orthopedic shoes	-
Behavioral problems	Anxious and aggressive behavior in unexpected situations	Aggressive toward sibling, often screams, runs away	Easily over-stimulated, repetitive movements	-	Acts young for age, frustration when he cannot pronounce words

From the educational background we can infer that children in this study had an IQ between 40 and 60, based on the eligibility for special education in the Netherlands.

**Table 2 animals-09-00716-t002:** Total time of spoken language measured in seconds and DCRP measures for the five cases.

Participants	Seconds of Spoken Language Participant		Seconds of Spoken Language Therapist		RR		RR_peak_		Q_los_	
	First	Last	First	Last	First	Last	First	Last	First	Last
Pp 1	232	278	542	490	0.18	0.26	0.24	0.29	−0.03	−0.02
Pp 2	199	138.5	599	450	0.17	0.19	0.23	0.24	−0.07	−0.03
Pp 3	113.5	126	562.2	605.5	0.08	0.09	0.11	0.12	0.005	0.02
Pp 4	176	174.5	854	943	0.19	0.07	0.06	0.04	−0.02	−0.01
Pp 5	153	34.5	509	276.5	0.2	0.08	0.16	0.04	−0.04	0.04

DCRP: diagonal cross-recurrence profile; RR: recurrence rate; RR_peak_: the highest proportion of recurrence; Q_los_: the proportion of matching behavior on the left side of the LOS (Line of Synchrony) divided by the proportion of matching behavior on the right side.

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
