# Peer review of "Verbal Interactional Synchronization between Therapist and Children with Autism Spectrum Disorder during Dolphin Assisted Therapy: Five Case Studies"

_animals, 2019, doi:10.3390/ani9100716_

Round 1

Reviewer 1 Report

I’m not sure if Animals is the appropriate journal for this research.  While the therapeutic intervention used dolphins, the research design does not allow for an investigation of the impact of the animal on the outcomes.  There are also no suggestions for future research to determine the impact, if any, of the animal.  I realize in the case study design, the participant serves as their own control; yet, I am left to wonder if the results would have been the same if the therapist had been working alone with the client, if there had been a human only addition, if a different animal had been added?  This brings to mind the construct validity issue raised by Marino (Lori Marino (2012) Construct Validity of Animal-Assisted Therapy and Activities: How Important Is the Animal in AAT?, Anthrozoös, 25:sup1, s139-s151, DOI: 10.2752/175303712X13353430377219).

Several sections, I would suggest grammatical changes of “to verb” converted to verb + ing

Page 2, line 47 “to achieve” would then be “in achieving”

Page 7, line 236 “to produce” would then be “producing”

Page 7, lines 236-237 “to express” would then be “expressing”

Page 2 Nice job creating a sense of importance and urgency to the problem and research

Page 4, lines 146-150 – are the raters blind to whether they are viewing the first or final session?  Is it possible to be blind to this?  Given the results and that all children, but especially those with developmental challenges can have “bad days”, why weren’t additional sessions coded?  What if the last session was atypical?  Why not code a middle session as well?

Page 5, line 176 – I don’t think it is the reader’s job to find an earlier article to understand what you are doing here.  I appreciate the reference to the earlier article, should I want additional information.  But I felt that so little information on this procedure was provided that I am very unclear on what is done or what is meant by the statistics and the graphs.  Consider a brief explanation  If such an explanation disrupts the flow of the narrative, add it as an appendix. 

Page 5, Figure 1- why are there boxes in the middle of the figure?  I can’t read the text under the line in the graph.  I just figured this out…..if viewed online, these are left and right pointing arrows, but if printed they are boxes superimposed on the words.  Consider trying to insert this information as a picture so it does not change when printed.  People will view your final article online and will also print it.

Figures 2 through 6 – why are the x and y axes not labeled in these figures but they are labelled in Figure 1?

Figures 2 through 6 – when I view online, I can easily see the gray for Session 1 and black for Final Session.  But when I printed there was no distinction – both session lines were black.  Consider using a stronger color contrast or dashed line (not sure this is possible) in case future readers print the article so they can distinguish the two different times.

Page 11, line 384 – You state When the number and length of the participants’ verbal expressions increases, the therapist may respond to this by using more complex sentences as well, which may take more time to process for the participant, thereby decreasing adequate turn-taking.” Can’t you determine this?  Why do you need to say “may…use more complex sentences”?  Why can’t you measure the length of the therapist’s sentences and those of the participant?

Page 11, line 391 – Similarly, you state Another reason for the discrepancy between the qualitative and the quantitative data could be that improved verbal skills of the children lead to longer turns, which ultimately results in a lower total number of turns.“ Again, can’t you determine this?  Why do you need to say “could …lead to longer turns….lower total number of turns”?  Why can’t you measure the length and number of turns?

Page 11, lines 407 and 409 – shouldn’t “participant 1 and 2” be “participants 1 and 2”?

Strong conclusion and section on limitations with good ideas for future research in this area regarding initial verbal skill ability of participants.  Not sure if that would be measured with standardized tests or professional evaluation via observation or videotaping of participant prior to study.

Author Response

Reviewer 1

Thank you for your extensive review and your helpful comments/suggestions. We provide an answer to the issues you raised below. The line numbers refer to the revised manuscript.

I’m not sure if Animalsis the appropriate journal for this research.  While the therapeutic intervention used dolphins, the research design does not allow for an investigation of the impact of the animal on the outcomes. There are also no suggestions for future research to determine the impact, if any, of the animal.  I realize in the case study design, the participant serves as their own control; yet, I am left to wonder if the results would have been the same if the therapist had been working alone with the client, if there had been a human only addition, if a different animal had been added? This brings to mind the construct validity issue raised by Marino (Lori Marino (2012) Construct Validity of Animal-Assisted Therapy and Activities: How Important Is the Animal in AAT?, Anthrozoös25:sup1, s139-s151, DOI: 10.2752/175303712X13353430377219).

Thank you for your comment, we understand your question with regard to the impact of an animal in this research. We therefore added an explanation to the discussion:

“Because this exploratory study intended to investigate whether DAT influences changes in turn-taking behavior, it was beyond our scope to compare the effect of the therapy to a control group. However, in an earlier study (Griffioen and Enders, 2014) we reported the positive effect of this program on verbalization and social interaction measures, by comparing DAT to two control groups (using a radio-controlled boat in a swimming pool, and a waitlist). As advised by Marino (2012) weminimized construct confounding as we exposed the control group and the DAT intervention group to highly similar procedures”. (page 11, lines 341-347)

“In future studies, the effect of Animal-Assisted interventions on synchrony could be further investigated, using conditions of therapy with and without animals, or by comparing synchrony between child and therapist in the presence of different animals, such as dogs or horses”.  (page 13, lines 407-410)

Several sections, I would suggest grammatical changes of “to verb” converted to verb + ing

Thank you for your suggestion to change from to verb to verb + ing, we changed this accordingly

Page 2, line 47 “to achieve” would then be “in achieving”

Page 7, line 218 “to produce” would then be “producing”

Page 7, line 219 “to express” would then be “expressing”

Page 2 Nice job creating a sense of importance and urgency to the problem and research; Page 4, lines 146-150 – are the raters blind to whether they are viewing the first or final session?  Is it possible to be blind to this? 

Thank you for your question, we added this information about the procedure:

The raters were not blind to the session number. Note, however, that the categories required little to no interpretation of the coder. It was simply noting whether or not someone speaks. In addition, before coding the videos, students went through a training and a percentage of agreement of 80% with the expert rater was required. (see methods section 175-190).

Given the results and that all children, but especially those with developmental challenges can have “bad days”, why weren’t additional sessions coded?  What if the last session was atypical?  Why not code a middle session as well?

Thank you for your question. Given that this was an exploratory study (do we see any changes in synchrony?), and because of the labour-intensive nature of the coding, we did not code more sessions. We added this as a limitation to our discussion.

“Given that this was an exploratory study (do we see any changes in synchrony?), and because of the labour-intensive nature of the coding, this study only compares the child-therapist synchrony in two sessions. Future research could investigate more sessions to track the development of therapist-child synchrony over time”. (Page 12, Lines 387-391)

Page 5, line 176 – I don’t think it is the reader’s job to find an earlier article to understand what you are doing here.  I appreciate the reference to the earlier article, should I want additional information.  (Reference 60)But I felt that so little information on this procedure was provided that I am very unclear on what is done or what is meant by the statistics and the graphs. Consider a brief explanation.  If such an explanation disrupts the flow of the narrative, add it as an appendix. 

We thank the reviewer for this question and provided more information on this particular study.

“This study (60) investigated the course of behavioral synchrony on movement patterns (and the outcome on problem behaviors) of children with Down syndrome (DS) and children with ASD while interacting with a dog. We used CRQA to operationalize the movement synchrony between a dog and child and found an increase in the coupling between child and dog during the final session, meaning that child and dog became mutally attuned in their movements”. (page 5, lines 197-202)

Page 5, Figure 1- why are there boxes in the middle of the figure? 

I can’t read the text under the line in the graph.  I just figured this out…..if viewed online, these are left and right pointing arrows, but if printed they are boxes superimposed on the words.  Consider trying to insert this information as a picture so it does not change when printed.  People will view your final article online and will also print it. 

Figures 2 through 6 – why are the x and y axes not labeled in these figures but they are labelled in Figure 1? 

Figures 2 through 6 – when I view online, I can easily see the gray for Session 1 and black for Final Session.  But when I printed there was no distinction – both session lines were black.  Consider using a stronger color contrast or dashed line (not sure this is possible) in case future readers print the article so they can distinguish the two different times.

Thank you for your advice, this will improve readability of our paper. We changed the figures to PDF files for better viewing. In addition, we labeled all the figures X and Y axes. We also changed the grey lines into dashed lines for better viewing.

Page 11, line 384 – You state “When the number and length of the participants’ verbal expressions increases, the therapist may respond to this by using more complex sentences as well, which may take more time to process for the participant, thereby decreasing adequate turn-taking. Can’t you determine this?  Why do you need to say “may…use more complex sentences”?Why can’t you measure the length of the therapist’s sentences and those of the participant?  LINE 389

Thank you for your question and suggestion. We only coded start and end times of spoken language. As a result, we can only measure the duration of speech time. For three out of five children, the average duration of spoken language increased during the final session of DAT. However, we are hesitant to include this information in the results section or discussion, because we have not measured the complexity of the utterances, nor the number of words. The total time of spoken language measured in seconds were simply of longer  duration.

We added to the discussion:

“For three out of five children, the average duration of spoken language increased during the final session of DAT(see table 2) while the duration of spoken language of the therapist measured in seconds decreased”. (page 12, lines 363-365)

Page 11, line 391 – Similarly, you state “Another reason for the discrepancy between the qualitative and the quantitative data could be that improved verbal skills of the children lead to longer turns, which ultimately results in a lower total number of turns.“ Again, can’t you determine this?  Why do you need to say “could …lead to longer turns….lower total number of turns”?  Why can’t you measure the length and number of turns? LINE 397 future research.

Thank you for your remark, reviewer 4 advised to delete the qualitative descriptions and focus on the quantitative results solely.  As a result of this, we corrected the discussion and deleted the phrase mentioned above.

Page 11, lines 407 and 409 – shouldn’t “participant 1 and 2” be “participants 1 and 2”? Lines 355, 357

Thank you for your comment, indeed it should be plural. We changed this accordingly.

Strong conclusion and section on limitations with good ideas for future research in this area regarding initial verbal skill ability of participants.  Not sure if that would be measured with standardized tests or professional evaluation via observation or videotaping of participant prior to study.

Thank you for your comments. We think future research will benefit from using professional evaluation via observation or videotaping of participant prior to study and/or standarized verbal tests. However this will need extensive research and a pilot study to confirm. We added to the discussion: “e.g. using videotapes of the particpants prior to the study” (page 12, line 376-377).

Reviewer 2 Report

I know the researchers specifically commented on the dolphin trainer being present to monitor signs of distress in the dolphin. I also noticed in the discussion that different dolphins were used on different occasions, depending on condition and availability. I think it needs clarifying if any of the dolphins were distressed by this study, and if so, the implications for animal welfare.

Author Response

Reviewer 2

I know the researchers specifically commented on the dolphin trainer being present to monitor signs of distress in the dolphin. I also noticed in the discussion that different dolphins were used on different occasions, depending on condition and availability. I think it needs clarifying if any of the dolphins were distressed by this study, and if so, the implications for animal welfare.

Thank you for your suggestion. We provided more clarification on the procedure and the implications for animal welfare in the methods section.

“We worked with 3 dolphins for 5 children, and in principle each child worked with the same dolphin during all sessions. At the start of a session, the dolphin trainer would call the dolphins to the raft, and one dolphin was chosen to work with during the session. After the first session, the preferred choice of dolphin was always the dolphin the child had worked with before. Due to pregnancy of one dolphin we had to change dolphins for two children in the last session. In addition, because the dolphins were not forced to interact with the children, we only once switched dolphins when the regular dolphin did not come to the raft when the dolphin trainer called.  We did not have to cancel a session due to stress signals of the dolphins. In an earlier study the stress signals of the dolphins during a training session, a therapy session of this DAT program, and at rest time were investigated (Boneh and & Leeuwen, 2003; Griffioen & Enders, 2014). In this study, the respiratory frequency of the dolphins was measured as a parameter of arousal. Similar levels of respiratory frequency were measured during days with and without training sessions”. (Page 4, Lines 146-157)

Reviewer 3 Report

The authors present five cases of a therapeutic interaction with children assisted by dolphins.  To take the turn-taking as central measure of communication analysis for synchronization sounds very well. The application of recurrence analyses is an appropriate and interesting approach.

There were no statistical or scientific hypotheses formulated. The experimental plan was diffuse (different dolphins per session, session plan). The limitations were listed by the authors themselves. However, the applied method of analysis was described very detailed. The literature list is enormous about that. Generally the number of references is worth to belong to a textbook or at least theses, not to a (5) case report.

It was not given, where the therapy took place, open water, basin/pool, even country or location, leading organization, where the therapists received their (which kind of?) education/training. The usual living conditions of the animals were not described. The general concept of the therapy was even not mentioned. How parents had access to the therapy for their children? This is fundamental information for case discussions.

Over-interpretation of casuistic results, 5 cases three different result patterns. No recommendation was made how these results could be useful in future therapy sessions but large ambiguous theoretical discussions were presented.

It was my final impression, that occasional data were used for gambling around with impressive methods. I could not recognize any real interest in children or therapy effect or animal wellbeing. The only aim of the manuscript seems to be, to present oneself being able to use the analysis methods as an end in itself.

Minor remarks

The numerical results cannot be recognized from the graphs. They should have been presented as a summarizing table for all cases to allow any summary.

Author Response

Reviewer 3

The authors present five cases of a therapeutic interaction with children assisted by dolphins.  To take the turn-taking as central measure of communication analysis for synchronization sounds very well. The application of recurrence analyses is an appropriate and interesting approach. There were no statistical or scientific hypotheses formulated. The experimental plan was diffuse (different dolphins per session, session plan).

The limitations were listed by the authors themselves. However, the applied method of analysis was described very detailed. The literature list is enormous about that. Generally the number of references is worth to belong to a textbook or at least theses, not to a (5) case report. 

It was not given, where the therapy took place, open water, basin/pool, even country or location, leading organization, where the therapists received their (which kind of?) education/training. The usual living conditions of the animals were not described.

Response: Thank you for your remarks. In response, we have updated the information in the method section:

“The therapy took place in the Netherlands at the Dolphinarium Harderwijk. The dolphins live in a artificial lagune of 7000 square meter and are not used for shows. They live together as a small pod (17 dolphins, including males, females, juveniles and babies). The SAM foundation was responsible for the organization of the program. They provided the trained speech and language therapists. All therapist were trained at universities of applied science in the Netherlands and had followed an intensive training on how to work with the dolphins, organized by the foundation”. (Page 4, Lines 125-130)

“We worked with 3 dolphins for 5 children, and each child worked with the same dolphin during all sessions. At the start of a session, the dolphin trainer would call the dolphins to the raft, and one dolphin was chosen to work with during the session. After the first session, the preferred choice of dolphin was always the dolphin the child had worked with before. Due to pregnancy of one dolphin we had to change dolphins for two children in the last session. In addition, because the dolphins were not forced to interact with the children, we only once switched dolphins when the regular dolphin did not come to the raft when the dolphin trainer called.  We did not have to cancel a session due to stress signals of the dolphins. In an earlier study the stress signals of the dolphins during a training session, a therapy session of this DAT program, and at rest time were investigated (Boneh and & Leeuwen, 2003; Griffioen & Enders, 2014). In this study, the respiratory frequency of the dolphins was measured as a parameter of arousal. Similar levels of respiratory frequency were measured during days with and without training sessions”. (Page 4, Lines 146-157)

The general concept of the therapy was even not mentioned. How parents had access to the therapy for their children? This is fundamental information for case discussions.

Response: Thank you for your comment, we agree this information was lacking and we did provide the neccesary information of the general concept of the therapy in the methods section:

“Parents voluntarily registered their children for therapy after reading newspaper articles about this program. Upon registration, parents were asked for permission to participate in the study. After registration children were placed on a waitlist and were called in for an extensive intake as soon as there was room to start therapy. The exclusion criteria for participation included fear of water, epilepsy, poor vision and a history of aggressive behavior toward animals. The therapy program was partly sponsored and therefore parents paid a reduced price.” (Page 3, Lines 108-114)

Over-interpretation of casuistic results, 5 cases three different result patterns. No recommendation was made how these results could be useful in future therapy sessions but large ambiguous theoretical discussions were presented.It was my final impression, that occasional data were used for gambling around with impressive methods. I could not recognize any real interest inchildren or therapy effect or animal wellbeing. The only aim of the manuscript seems to be, to present oneself being able to use the analysis methods as an end in itself.

Response: The reviewer is right that cross-recurrence quantification analysis (CRQA) might be a rather complicated method. The advantage of this method, however, is that it can detect synchrony in conversations (in this case turn-taking dynamics) across all possible timescales, ranging from seconds to the duration of the entire interaction. This means that even when the appropriate turn is (slightly) delayed, it will still be detected. (see also methods section, line 202-205).

With regard to the interest in children therapy or animal well being: we hope that by adding more information about therapy protocol, recruitment and the absence of stress signals, we have sufficiently addressed your concern.

Minor remarks

The numerical results cannot be recognized from the graphs. They should have been presented as a summarizing table for all cases to allow any summary.

Thank you for your suggestion. We added Table 2 to the results section, in which descriptive statistics related to the CRQA analysis are reported.

Table 2: Total time of spoken language measured in seconds and DCRP measures for the five cases and the sessions (first and last)

seconds of spoken language participant

seconds of spoken language therapist

RR

RRpeak

Qlos

first

last

first

last

first

last

first

last

first

last

Pp 1

232

278

542

490

.18

.26

.24

.29

-.03

-.02

Pp 2

199

138.5

599

450

.17

.19

.23

.24

-.07

-.03

Pp 3

113.5

126

562.2

605.5

.08

.09

.11

.12

.005

.02

Pp 4

176

174.5

854

943

.19

.07

.06

.04

-.02

-.01

Pp 5

153

34.5

509

276.5

.2

.08

.16

.04

-.04

.04

(Page 6, Lines 233-236)

Reviewer 4 Report

This paper looks at “dolphin therapy” and its effects on turn-taking in children diagnosed with autism. The paper concludes with the assertion that turn-taking has been improved, but is most evident in children with stronger verbal communication at the start of the study.

I’d like to begin my review by noting two general points which are important context for the rest of my review. I’d encourage the authors (and particularly the editors) to bare in mind this context when considering my other points which are detailed below.

So, first: I’m a sociologist who works overwhelmingly with qualitative data forms and, accordingly, I do not feel able to comment in any depth on the suitability of the quantitative tests chosen or the quality of the analysis. From this (unenlightened) vantage point I do have lots of questions about the nature of the analyses: (1) Why are there no inferential statistics presented? (2) Why are the children analysed individually, rather than as a group? (3) Why is there no control group so that we can get a sense of what these numbers actually mean? (4) Why are sessions 2-5 excluded from the analyses so we can see if a trend develops over time? (5) Why does the between-participant variation dwarf the between-session variation (generally speaking) and does this matter? (6) Is the fact that the behaviour of the dolphin – indeed, the *actual dolphin* - differs between sessions such an *enormous* variable that it makes other analyses kind of mute? My initial impression is that the conclusion that “The most important result revealed in this study, that is, turn-taking during DAT seems to increase for children with reasonable verbal communication skills” (L416-417), a statement emphasised in both the lay summary and the abstract, is not a sustainable conclusion given the limited data set and huge inter-participant variation. But – BUT – I absolutely defer to a reviewer with better knowledge of the methodology under question and would happily disregard these concerns.

Second: It is assumed throughout the article that ‘reduced autistic symptoms’ is a good thing but this is an assertion which is, as I’m sure the authors know, frequently challenged by members of the autistic community. Numerous autistic individuals and scholars (Damien Milton and Mel Yergeau are names which immediately come to mind) would, I am sure, contest the premise that asynchronised turntaking is a problem with the autistic individual that needs to be fixed. Regardless of what one makes of these arguments, I’m of the personal belief that we’re at a point now where it isn’t ethical to ignore these voices and that writing as if it is beyond doubt that autism is a problem to be fixed is questionable. Still, I understand that that this is far from a universally accepted position and leave it to the editors and authors to consider the best way to advance.

So, after that prelude I’ll get on with the nitty gritty of the qualitative analysis. The paper is very clearly written and very well structured, it's easy to read and the central message is clear. That said, the qualitative portions of the paper are a bit of an afterthought and I’d encourage the authors to either leave them out entirely or to radically rework them – if the basis of the paper was simply these analyses then there would be no option other than to recommend reject. 

The problem is multi-fold.  First, much of qualitative analysis is not actually qualitative and still relates to either quantitative data (e.g. “During the last session of DAT, the participant’s turns are of a longer duration” (L211); “The qualitative descriptions show that all participants seemed to improve in their turn-taking skills and their formulation of sentences (e.g., from 1 to 2 words sentences to 4 to 5 word sentences)” (L360) or quantifiable phenomena which are already quantifiable given the current data set (“In these questions he seems to use more words in a sentence, compared to the first session” – L332; the therapists use of complex sentences (L384) or participants have longer turns in the final session (L392) as possible explanations for non-significant effects). It would, I assume, be easy enough to quantify these data but, regardless, it isn’t qualitative analysis. Second, there are assertions and suppositions made about, e.g. intentionality (e.g. "Done!" (which could mean he wants to stop the activity) (L275); “She seems to choose her words carefully” (204)) which simply can’t be evidenced given the data presented. We require thick description if we are to make such judgements.

I am of the belief that the central problem here is a lack of analytic framework comparable to that used for quantitative analysis. There are, of course, many very well-established modes of qualitative data analysis but, given the micro-sociological focus here I do find the decision not to utilize conversation analysis – which is both specifically designed to examine turn-taking and, as a recent special edition of JADD demonstrated, perfectly suited to the study of interactions with autistic individuals – an odd one. Undertaking conversation analysis is hard and requires significant expertise. This is, however, also true of quantitative analysis and there is no reason to assume that qualitative analysis is or should be intrinsically easier or less rigorous.  Regardless of the frame, some attempt to impart a rigorous approach to qualitative analysis is much needed. I've recommended reject on the basis of the qualitative analysis but, if another reviewer felt that the quantitative portions of the paper were sound and important then I think it would be perfectly acceptable to go with that conclusion given the dominance of quantitative analysis. 

Small, additional points:

Section 2.3 is exceptionally clearly written but there is a lot going on which, to someone not familiar with the methodology seems non-obvious: why were the children only allowed to gesture after getting permission? Is there evidence this works? Why are the emotion boards present and then thrown in the water? Is there evidence this works? To be clear – I’m absolutely not questioning this methodology, just saying that a line or two of supporting information would be ideal. As noted above, the paucity of information regarding qualitative analysis is problematic.

L64 – “Therapies that increase 64 synchrony but at the same time create a positive non-demanding environment are therefore crucial.” This seems like a logical conclusion based upon the (relatively small) literature cited above, but I don’t think this can be claimed as beyond doubt, based upon what’s written here. Suggest a re-phrasing.

L66 – “Some authors argue that such a positive and non-demanding environment can be achieved by 66 incorporating animals into therapy.” Please see Roslyn Malcolm’s recently published work on explanations for why animal therapies work.

References in review

Malcolm, R., Ecks, S. and Pickersgill, M., 2018. ‘It just opens up their world’: autism, empathy, and the therapeutic effects of equine interactions. Anthropology & medicine25(2), pp.220-234.

Milton, D.E., 2012. On the ontological status of autism: the ‘double empathy problem’. Disability & Society27(6), pp.883-887.

O’Reilly, M., Lester, J.N. & Muskett, T. J Autism Dev Disord (2016) 46: 355. https://doi.org/10.1007/s10803-015-2665-5

Yergeau, M., 2018. Authoring autism: On rhetoric and neurological queerness. Duke University Press.

Author Response

Reviewer 4

Thank you for your extensive review and your helpful comments/suggestions. We provide an answer to the issues you raised below. The line numbers refer to the revised manuscript.

First, questions about the nature of the analyses.

1.1 Why are there no inferential statistics presented?

Since we are not working with a large sample, and since we are not deriving estimates of a larger population, we chose not to analyze the data using inferential statistics (t-tests, Anovas, etc). However, in response to your comment, we added Table 2 to the results section, in which descriptive statistics related to the CRQA analysis are reported. Page 6, lines 233-236

Table 2: Total time of spoken language measured in seconds and DCRP measures for the five cases.

seconds of spoken language participant

seconds of spoken language therapist

RR

RRpeak

Qlos

first

last

first

last

first

last

first

last

first

last

Pp 1

232

278

542

490

.18

.26

.24

.29

-.03

-.02

Pp 2

199

138.5

599

450

.17

.19

.23

.24

-.07

-.03

Pp 3

113.5

126

562.2

605.5

.08

.09

.11

.12

.005

.02

Pp 4

176

174.5

854

943

.19

.07

.06

.04

-.02

-.01

Pp 5

153

34.5

509

276.5

.2

.08

.16

.04

-.04

.04

1.2 Why are the children analysed individually, rather than as a group?

This study was set out to explore the shared interactional dynamics between therapist and child during dolphin-assisted therapy. We illustrate these dynamics in five cases. Given the small number of participants, and given that the aim of the study was to examine the underlying therapist-child dynamics, we chose to conduct a case study.

1.3 Why is there no control group so that we can get a sense of what these numbers actually mean? 

“Because this exploratory study intended to investigate whether DAT influences changes in turn-taking behavior, it was beyond our scope to compare the effect of the therapy to a control group. However, in an earlier study (Griffioen and Enders, 2014) we reported the positive effect of this program on verbalization and social interaction measures, by comparing DAT to two control groups (using a radio-controlled boat in a swimming pool, and a waitlist). As advised by Marino (2012) weminimized construct confounding as we exposed the control group and the DAT intervention group to highly similar procedures”. (page 101 lines 341-347).

1.4 Why are sessions 2-5 excluded from the analyses so we can see if a trend develops over time?

While focusing on a trend would indeed be interesting, we first and foremost wanted to know if children’s interactional synchrony (i.e. turn-taking) would indeed grow over time during the therapy program, thereby comparing their turn-taking between the first and last session.  

1.5 Why does the between-participant variation dwarf the between-session variation (generally speaking) and does this matter?

We are not sure what the reviewer means in this regard, our apologies. We compared therapist-participant couples to themselves (that is, their turn-taking in the first and last session), and we did not conduct inferential statistics (e.g., Anova) to test group effects, therefore no between-participant variation was calculated or used in statistical tests.

1.6 Is the fact that the behaviour of the dolphin – indeed, the *actual dolphin* - differs between sessions such an *enormous* variable that it makes other analyses kind of mute?

The dolphin showed similar behavior in each session. That is, children were taught to use gestures to make the dolphin perform certain actions (e.g., jump, or lie on its back so that the child could stroke the dolphin), and the dolphin performed these actions. The conversation between the therapist and child around the dolphin’s activities was input for our analyses.

My initial impression is that the conclusion that “The most important result revealed in this study, that is, turn-taking during DAT seems to increase for children with reasonable verbal communication skills” (L416-417), a statement emphasised in both the lay summary and the abstract, is not a sustainable conclusion given the limited data set and huge inter-participant variation. But – BUT – I absolutely defer to a reviewer with better knowledge of the methodology under question and would happily disregard these concerns.

The reviewer is right that our most important conclusion is that turn-taking during DAT increased for children with reasonable verbal communication skills. We are aware of the limited possibilities to generalize our findings, given that we presented five case studies. In the discussion we therefore note (Page 12, Lines 380-383): “The most important result revealed in this study, that is, turn-taking during DAT seems to increase for children with reasonable verbal communication skills, could be further examined in larger studies, that could also include other variables, such as the age of the children, the severity of the disorder, or gender.”

Second, it is assumed throughout the article that ‘reduced autistic symptoms’ is a good thing but this is an assertion which is, as I’m sure the authors know, frequently challenged by members of the autistic community. Numerous autistic individuals and scholars (Damien Milton and Mel Yergeau are names which immediately come to mind) would, I am sure, contest the premise that asynchronised turntaking is a problem with the autistic individual that needs to be fixed. Regardless of what one makes of these arguments, I’m of the personal belief that we’re at a point now where it isn’t ethical to ignore these voices and that writing as if it is beyond doubt that autism is a problem to be fixed is questionable. Still, I understand that that this is far from a universally accepted position and leave it to the editors and authors to consider the best way to advance.

Thank you for your thoughtful comment. It was by no means our intention to marginalize the autistic community. We therefore re-read the paper and chose our words more carefully on:

Line 17 (abstract): increased social skills Line 56: increased social skills Line 77: removed the words “autistic symptoms” Line 80: improved social interaction Comments/critique about the qualitative analysis

We thank the reviewer for the extensive comments about this part of the paper. The goal of our qualitative descriptions was not to provide a full qualitative analysis (which is why we referred to this part of the results as “qualitative description”), but to provide a context for the reader and a clear description of the situation and turn-taking during the interaction. We fully agree with the reviewer that in terms of a qualitative analysis this paper falls short. The reviewer suggests leaving the qualitative descriptions out entirely, and we followed this advice in the revised paper.

Section 2.3 is exceptionally clearly written but there is a lot going on which, to someone not familiar with the methodology seems non-obvious: why were the children only allowed to gesture after getting permission? Is there evidence this works? Why are the emotion boards present and then thrown in the water? Is there evidence this works? To be clear – I’m absolutely not questioning this methodology, just saying that a line or two of supporting information would be ideal.

Thank you for your suggestion to add extra information about the procedure. There are two reasons why children had to ask for permission before gesturing to the dolphin. First, as mentioned in the text (Page 4, Line 176), in this way children’s interactions with the therapist were reinforced. Second, if children start gesturing toward the dolphin without notice, this could possibly lead to a confusing situation for the dolphin (e.g., when they use several gestures in a row, accidently use a wrong gesture, gesture while the dolphin trainer briefly attends to the dolphin, etc.). We therefore added to the sentence on line 177: “and no confusing situations for the dolphin would occur”

In addition, we added the following explanation on the emotion boards:

“Since recognizing, naming and describing emotions is something children with ASD often have trouble with, the therapy program incorporated a task in which these skills are practiced. To include the dolphin in this task, the boards were thrown into the water for the dolphin to return.” (Page 5, Lines 174-177)

L64 – “Therapies that increase synchrony but at the same time create a positive non-demanding environment are therefore crucial.” This seems like a logical conclusion based upon the (relatively small) literature cited above, but I don’t think this can be claimed as beyond doubt, based upon what’s written here. Suggest a re-phrasing.

Our wording may indeed be a little too strong here. We changed this into: “are therefore thought to be beneficial” on line 18 (abstract), 66, 82 and 326.

however a positive non-demanding environment is therefore thought to be beneficial.

Therapies that increase synchrony but at the same time create a positive non-demanding environment are therefore thought to be beneficial.

Therapies that aim to increase synchrony are therefore valuable, but a positive non-demanding environment seems therefore thought to be beneficial.(30).

A positive non-demanding environment is therefore thought to be beneficialto increase synchronizing verbal behavior, such as turn-taking,

L66 – “Some authors argue that such a positive and non-demanding environment can be achieved by incorporating animals into therapy.” Please see Roslyn Malcolm’s recently published work on explanations for why animal therapies work.

Thank you for your suggestion, we added Malcolm’s work to line 68.

“Some authors argue that such a positive and non-demanding environment can be achieved by incorporating animals into therapy, and that animals would contribute to children’s social attunement (69).

Reference list:

Malcolm, R., Ecks, S. and Pickersgill, M., 2018. ‘It just opens up their world’: autism, empathy, and the therapeutic effects of equine interactions. Anthropology & medicine25(2), pp.220-234.

Round 2

Reviewer 3 Report

The manuscript was significantly improved. As an exploratory study it looks now well presented. I would like to know more about the stress assessment in the dolphins. How respiration was measured? There was also no physiological stress (arousal) assessment of the kids discussed (there are literature findings).

Author Response

Thank you for your final comments:

1. The manuscript was significantly improved. As an exploratory study it looks now well presented. I would like to know more about the stress assessment in the dolphins. How respiration was measured?

We expanded the sentences about the stress signals of the dolphins:

"We did not have to cancel a session due to stress signals of the dolphins. In an earlier study the stress signals of the dolphins were observed before, during and after therapy sessions (Boneh and & Leeuwen, 2003; Griffioen & Enders, 2014). In this study, the respiratory frequency of the dolphins was measured as a parameter of arousal.  Observations were made by means of a digital camera and by an observer, who noted the breathing frequency and sounds of the dolphins. The videotape was watched afterward and all behaviors were recorded in an ethogram. No significant differences between therapy sessions and control sessions were found."(lines 153-159)

2. There was also no physiological stress (arousal) assessment of the kids discussed.

All therapy sessions were watched by the parents and evaluated afterward. Parents reported no signs of stress signals. In addition, the student observers did not report any signs of stress in the children’s behavior. All parents evaluated the therapy sessions as positive at the end of the program.